# Structure of Genes Encoding Oxidosqualene Cyclases—Key Enzymes of Triterpenoid Biosynthesis from Sea Cucumber *Eupentacta fraudatrix*

**DOI:** 10.3390/ijms252312881

**Published:** 2024-11-29

**Authors:** Sergey N. Baldaev, Viktoria E. Chausova, Ksenia V. Isaeva, Alexey V. Boyko, Valentin A. Stonik, Marina P. Isaeva

**Affiliations:** 1G.B. Elyakov Pacific Institute of Bioorganic Chemistry, Far Eastern Branch, Russian Academy of Sciences, 159, Pr. 100 let Vladivostoku, 690022 Vladivostok, Russia; baldaevsergey@gmail.com (S.N.B.); v.chausova@gmail.com (V.E.C.); issaevakssenia@gmail.com (K.V.I.); stonik@piboc.dvo.ru (V.A.S.); 2Institute of High Technology and Advanced Materials, Far Eastern Federal University, Ajax Bay 10, Russky Island, 690922 Vladivostok, Russia; 3A.V. Zhirmunsky National Scientific Center of Marine Biology, Far Eastern Branch, Russian Academy of Sciences, Palchevskogo Street 17, 690041 Vladivostok, Russia; alteroldis@gmail.com

**Keywords:** *Eupentacta fraudatrix*, 2,3-oxydosqualene cyclase, molecular docking, gene determination and analysis, phylogeny

## Abstract

Oxidosqualene cyclases (OSCs) are enzymes responsible for converting linear triterpenes into tetracyclic ones, which are known as precursors of other important and bioactive metabolites. Two OSCs genes encoding parkeol synthase and lanostadienol synthase have been found in representatives of the genera *Apostichopus* and *Stichopus* (family Stichopodidae, order Synallactida). As a limited number of sea cucumber OSCs have been studied thus far, OSCs encoding gene(s) of the sea cucumber *Eupentacta fraudatrix* (family Sclerodactylidae, order Dendrochirotida) were investigated to fill this gap. Here, we employed RACEs, molecular cloning, and Oxford Nanopore Technologies to identify candidate OSC mRNAs and genes. The assembled cDNAs were 2409 bp (*OSC1*) and 3263 bp (*OSC2*), which shared the same CDS size of 2163 bp encoding a 721-amino-acid protein. The *E*. *fraudatrix* OSC1 and OSC2 had higher sequence identity similarity to each other (77.5%) than to other holothurian OSCs (64.7–71.0%). According to the sequence and molecular docking analyses, OSC1 with L436 is predicted to be parkeol synthase, while OSC2 with Q439 is predicted to be lanostadienol synthase. Based on the phylogenetic analysis, *E. fraudatrix* OSCs cDNAs clustered with other holothurian OSCs, forming the isolated branch. As a result of gene analysis, the high polymorphism and larger size of the OSC1 gene suggest that this gene may be an ancestor of the OSC2 gene. These results imply that the *E. fraudatrix* genome contains two OSC genes whose evolutionary pathways are different from those of the OSC genes in Stichopodidae.

## 1. Introduction

The far eastern sea cucumber *Eupentacta* (*Cucumaria*) *fraudatrix* is a producer of promising biologically active substances, in particular, a series of triterpenoid glycosides (saponins, the so-called cucumariosides) [1], exhibiting ichthyotoxic, antitumor, hemolytic, antifungal, and other pharmacological effects [2,3,4,5]. It has been found that in the sea cucumber *E. fraudatrix,* all free sterols result from de novo biosynthesis, and are mainly of the parkeol type with the Δ^9(11)^ bond, while saponins are predominantly formed from the lanosta-7,24-dienol (lanostadienol) type with the Δ^7(8)^ bond in the triterpene scaffolds [6]. In addition to their membrane-stabilizing function, unusual sterols of *E. fraudatrix* were found to reduce the membranolytic action of triterpenoid glycosides [6,7]. A total of 54 glycosides (26 sulfated, 18 nonsulfated, and 10 disulfated compounds) were isolated and detected from *E. fraudatrix* [8]. The in vitro study of biological activity of some cucumariosides from *E. fraudatrix* showed cytotoxic activity against Ehrlich carcinoma cells as well as mouse spleen lymphocytes [5,9]. In addition, cucumarioside I2, isolated from the sea cucumber, has a potent immunomodulatory effect at nanomolar concentration, enhancing anticancer response by polarizing mouse macrophages toward an M1-like phenotype [2,10]. This makes holothurian triterpene glycosides, as well as their biosynthesis, a promising area of research in biomedicine.

To date, the metabolic pathways of triterpenoid biosynthesis have been partially characterized in a limited number of sea cucumber (holothurian) species [11,12,13]. The biosynthetic precursor of triterpenes, sterols, and triterpene glycosides is 2,3-oxidosqualene, which is cyclized by a membrane oxidosqualene cyclase (OSC; EC 5.4.99.7). Among representatives of echinoderms such as sea stars (starfish) and sea urchins, one type of OSC (named lanosterol synthase, LSS) is encountered, catalyzing the cyclization of 2,3-oxidosqualene into lanosterol. In sea cucumbers, two OSC genes have been reported in four species: *Apostichopus japonicus*, *Stichopus horrensis, Stichopus chloronotus,* and *Parastichopus parvimensis* (now *Apostichopus parvimensis*) [12,13,14]. Based on its expression in lanosterol-deficient yeasts and the conversion of 2,3-oxidosqualene into parkeol or lanostadienol, these genes were named parkeol synthases (PSs) and lanostadienol synthases (LDSs), correspondingly [12]. Phylogenetic analysis revealed the PSs and LDSs grouped together, forming two distinct clades on the isolated branch, which stands alone from other animal OSCs. This suggests the evolution of two divergent *OSC*s from an ancestral sea cucumber *LSS* by gene duplication and neofunctionalization [12].

The previously published data make an important contribution to the understanding of triterpenoid biosynthesis in sea cucumbers, concerning only the family Stichopodidae (order Synallactida). Herein, we present the results of the transcript and gene structures determination of two *OSC*s from *E. fraudatrix*, whose deduced amino acid sequences showed notable differences when compared with OSCs from Stichopodidae. The molecular modeling and docking analyses revealed that OSC1 might be parkeol synthase and OSC2 might be lanostadienol synthase. These findings, firstly acquired for the family Sclerodactylidae (order Dendrochirotida), will bring some additional clarity to the origin and biosynthesis of triterpenoid glycosides in sea cucumbers.

## 2. Results and Discussion

### 2.1. E. fraudatrix Genome Codes Two Different OSCs

#### 2.1.1. cDNA Determination of OSC mRNAs

To design gene-specific OSC primers for RACE reactions, the draft transcriptome of *E. fraudatrix* [15] was used for a BLASTX search with the human LSS sequence as a query. The strategy for the determination of OSC encoding transcripts is presented in Appendix A. As a result of a series of 5′- and 3′-RACE reactions, different RACE products were obtained, depending on tissues of *E. fraudatrix* cDNA libraries; the 5′-fragments were approximately 600 bp (body wall) and 400 bp (gut) and the 3′-fragments were approximately 600 bp (body wall) and 1000 bp (gut). This means that at least two OSC isoforms (OSC1 and OSC2) could be transcribed from the sea cucumber *E. fraudatrix* genome. The RACE sequences were further used to design primers flanking OSC1 CDS (pair primers OSC1_start/OSC1_stop) and OSC2 CDS (pair primers OSC2_start/OSC2_stop). After amplification, cloning, and sequencing, the OSC1 and OSC2 CDSs derived from the body wall and gut cDNA libraries had the same size of 2163 bp. It is also worth noting that the full-length OSC1 transcript was obtained only from body wall tissue cDNA, whereas the OSC2 transcript was obtained only from intestinal cDNA (Appendix A).

In total, after assemblies of CDS and RACE fragments, the full-length OSC1 and OSC2 cDNAs (open reading frame, ORF) in addition to the CDS include 5′-untranslated regions (UTRs) of at least 105 bp and 299 bp, and 3′-UTRs of at least 141 bp and 801 bp, respectively. The OSC1 and OSC2 mRNAs encode a 721-amino-acid protein with calculated molecular weights of 82.531 and 82.502 kDa, respectively. Importantly, the predicted 5′UTRs of OSCs genes derived from a draft *E. fraudatrix* genome were validated with presented RACE-PCR results, where both 5′-UTRs started with G at position +1 and the first initial AUG codons with a different Kozak consensus sequence of -9GTGAGAGCG-1 for OSC1 mRNA and -9CTGCAAGGA-1 for OSC2 mRNA. Interestingly, the 5′-ends of mRNAs contained multiple ATG codons; however, all but one translated in truncated proteins. However, the obtained 3′-UTRs sequences did not confirm the BREAKER prediction results, since Sanger sequencing reactions were interrupted by repeats and poly(T) regions. In the obtained *OSC1* 3′-UTR, a polyadenylation signal site was not detected, whereas in the *OSC2* 3′-UTR it was located at +3004AATAAA+3009. Comparative cDNA analysis showed that the 5′-UTRs and 3′-UTRs did not share any significant similarity; however, OSC1 and OSC2-encoding sequences shared 79.96% similarity.

#### 2.1.2. Sequence Analyses of OSC1 and OSC2 Proteins

Based on Interpro search results, the deduced amino acid sequences of *E. fraudatrix* OSC1 and OSC2 belong to terpene synthase (IPR002365, https://www.ebi.ac.uk, access on 21 May 2024), which combines the evolutionary-related lanosterol synthase (EC 5.4.99.7), cycloartenol synthase (EC 5.4.99.8), and hopene synthase (EC 5.4.99) [16]. In addition, BLASTP/CDD searches showed that the OSCs belong to the ISOPREN_C2_like superfamily, Class II terpene cyclases, including squalene cyclase and 2,3-oxidosqualene cyclase [17]. These enzymes are integral membrane proteins, which are responsible for converting linear triterpenes such as squalene or oxidosqualene into cyclic triterpenes such as hopene, lanosterol, or others.

A comparison of the primary structures of OSCs and LSSs from echinoderms (Table 1) showed that the *E. fraudatrix* OSCs sequences shared relatively high identity (65–71%) and similarity (79–83%) with other holothurian OSCs. They demonstrated less than 58% identity and 73% similarity to LSSs from the sea urchin and starfish, which might indicate evolutionary divergence in their sterol biosynthesis pathways. The primary structures of holothurian OSCs had much lower identity (53–57%) and similarity (69–73%) to human LSS.

A comparison of the *E. fraudatrix* OSCs sequences with those of OSCs from sea cucumbers, sea urchin, starfish, and human was carried out by the multiple alignment of amino acid sequences (Figure 1). Like other OSCs, *E. fraudatrix* OSC1 and OSC2 contain the major functional domains (Figure 1): four or five QW motifs to stabilize the enzyme structure [18,19], a highly conserved DTTAE domain to bind and protonate a substrate [20,21,22,23], and an LWIHCR domain to provide the cyclization of a substrate [24,25]. Interestingly, the DTTAE motif is conserved in both *E. fraudatrix* OSCs, whereas in *Apostichopus* and *Stichopus Aj*LAS1, *Ap*PS and *Sh*OSC1 have DTTAE, and *Aj*LAS2, *Ap*LDS, and *Sh*OSC2 have DTSAE. Despite the established difference in catalytic function of *Aj*LAS1, *Ap*PS and *Aj*LAS2, *Ap*LDS [12,14], the substituted amino acids share similar properties, and these enzymes use the same substrate oxidosqualene. Moreover, the LSSs presented in Figure 1 have a DCTAE motif.

Notably, the sequence identity, and even the sequence similarity, alone did not help to identify the OSC orthologues (Table 1); *E. fraudatrix* OSC1 and OSC2 had higher sequence identity to each other (77.5%) than to *A. japonicus* LAS1 or LAS2 (67.3–70.6%). Because in our case the sequence identity between holothurian paralogues (*E. fraudatrix*, family Sclerodactylidae) is greater than that between holothurian orthologues (families Sclerodactylidae and Stichopodidae), it is very likely that the similarity of specific amino acid residues between the active sites should be more useful in identifying orthologues that perform a similar function by a definition. The active site of OSCs is mainly represented by highly conserved amino acid residues. However, superimposition of the *Hs*LSS, *Ap*LDS, and *Ap*PS homology models performed by Thimmappa et al. [12] revealed that the amino acid residue at position 444 alone can distinguish PSs (436L) and LDSs (444Q) from LSSs (444F). A comparison of the active sites showed that *Ef*OSCs do differ at this residue, resulting in the grouping of *Ef*OSC1 with *Aj*LAS1 (*Ap*PS) at 436L and *Ef*OSC2 with *Aj*LAS2 (*Ap*LDS) at 439Q (Figure 1).

### 2.2. Confirmation of Key L and Q Residues Distinguishing Parkeol and Lanostadienol Synthases by Molecular Docking

A comparative modeling approach was taken to infer the function and orthology of *E. fraudatrix* OSC1 and OSC2 from their structures and determine the specific amino acid residues required to form Δ^9-11^ or Δ^7-8^ bonds. Importantly, two homologues, *Aj*LAS1 and *Aj*LAS2, recently identified in the *A. japonicus* genome and characterized by yeast heterologous expression, showed that *Aj*LAS1 catalyzes the formation of the Δ^9-11^ bond (parkeol), whereas *Aj*LAS2 synthesizes lanostadienol with the Δ^7-8^ bond [12,13].

In addition, corresponding mutations were later introduced in *A. japonicus* PS (*Aj*LAS1) as well as in sea urchin *S. purpuratus* LSS, which verified the key role of leucine (436L) in the biosynthesis of parkeol and glutamine (444Q) in the biosynthesis of lanostadienol [12]. Namely, Gil77 yeast cells with *A. japonicus* PS^L436F^ and PS^L436Q^ mutants synthesized lanosterol and lanostadienol, respectively, in addition to parkeol. The Gil77 yeast cells with the mutant of *S. purpuratus* LSS^F440L^ showed detectable levels of parkeol, whereas Gil77 cells with the mutant of *S. purpuratus* LSS^F440Q^ synthesized detectable levels of lanostadienol [12].

To compare the putative active sites of *Ef*OSC1 and *Ef*OSC2 to those of the functionally characterized *Aj*LAS1 (parkeol) and *Aj*LAS2 (lanostadienol), their structural models were generated by homology modeling using human LSS (PDB 1W6K) as a prototype [26]. All models showed root-mean-square deviations of the Cα-atom of less than 1 Å (0.18 Å), indicating very similar overall structures to the prototype. The superposition of the obtained OSCs structures revealed TM-scores of 0.977 to 0.986 and RMSD values over all atoms of 0.928 to 1.392 Å (Appendix A). Modeling of enzyme–product docking with MOE S-scores lower than −11.5 identified pairwise differences in the binding sites; docking of parkeol into the *Ef*OSC1 or *Aj*LAS1, as well as lanostadienol into the *Ef*OSC2 or *Aj*LAS2, resulted in the local structural similarities despite the lack of overall sequence identities. The 2D structures of *Ef*OSCs and *Aj*LASs with active site residues within 5 Å around the products are shown in Appendix A. In contrast, docking of lanostadienol into the *Ef*OSC1 or *Aj*LAS1 and parkeol into the *Ef*OSC2 or *Aj*LAS2 led to similar patterns of unfavorable H-bonds (Appendix A).

Based on the resulting 3D models (Figure 2), the location of the residues of leucine (436L/435L) in *Ef*OSC1/*Aj*LAS1 and glutamine (439Q/444Q) in *Ef*OSC2/*Aj*LAS2, as well as threonine (448T/447T or 451T/456T), tyrosine (Y496/Y495 or Y498/Y503), and phenylalanine (F688/F690 or F691/F696) ones near the B ring of lanostadienol (Figure 2c,d) or C ring of parkeol (Figure 2a,b), is characterized by high structural similarity, suggesting that these particular residues, 436L and 439Q, may determine product specificity in the *Ef*OSCs.

Thus, based on the results of molecular docking, it can be inferred that in sea cucumber *E. fraudatrix,* OSC1 should cyclize 2,3-oxidosqualene into parkeol and be an orthologue of parkeol synthase *Aj*LAS1 (*Aj*PS), while OSC2 should cyclize it into lanostadienol and be an orthologue of lanostadienol synthase *Aj*LAS2 (*Aj*LDS). Further biochemical studies of the OSCs proteins and their mutants need to be carried out to prove the putative function of these enzymes.

### 2.3. E. fraudatrix OSC1 and OSC2 form a Phylogenetically Distinct Branch

To evaluate the evolutionary relationships of *E. fraudatrix* OSC1 and OSC2 in Echinodermata, phylogenetic trees were constructed using the ML method based on aligned amino acid sequences of known OSCs (PSs, LDSs, and LSSs) from Metazoa (Appendix A). Using OSCs from *Dictyostelium discoideum* (Amoebozoa, sister clade to Obazoa) and *Saccharomyces cerevisiae* (Holomycota, sister clade to Holozoa) as outgroups [27], OSCs from Metazoa formed two superclades (Figure 3). The first clade exclusively represented holothurian OSCs (*E. fraudatrix*, *A. japonicus*, *A. parvimensis,* and *S. horrens*), and the second clade included LSSs from Porifera, Placozoa, and Bilateria, as well as other echinoderms such as starfish and sea urchins. In the starfish and sea urchin genomes, a single OSC gene was identified, which, like in all other animals, encodes lanosterol synthase with F444. Based on our phylogenetic analysis, holothurian OSCs were grouped into clades according to their taxonomic positions; the first clade consisted of two paralogues, OSC1 and OSC2 from *E. fraudatrix* (family Sclerodactylidae), and the second clade consisted of two subclades corresponding to PSs and LDSs from members of the genera *Apostichopus* and *Stichopus* (family Stichopodidae) (Figure 3a). In case of exclusion of the holothurian OSCs from the alignment used for tree construction, the resulting tree reflected the taxonomic relationships inferred from their genome sequences (Figure 3b).

It is noteworthy that on the genomic tree [28], holothurians and sea urchins are grouped more closely, while sea stars are situated on the outer branch of Echinodermata. In contrast, in the OSC tree constructed, sea urchins clustered with sea stars, whereas holothurians formed an outer branch for the Metazoa clade with high branch node support. This suggests that despite the phylogenetic proximity of holothurians to other echinoderms, they demonstrate a distinct independent evolutionary lineage of OSCs driven by duplication events that provide the biosynthesis of unusual sterols and triterpenoid glycosides.

Together, despite the fact that *E. fraudatrix* OSCs formed a distinct branch rather than complementing orthologous groups of LDSs and PSs, we consider that the key residue in the position 444 (436L or 439Q) may determine not only the specific cyclization of the substrate, but also an attribution of the sequences to a certain orthologous group. The genetic distances observed between *E. fraudatrix* OSCs are significantly large, suggesting that their evolutionary divergence was driven by an acquired end-product specificity (in contrast to *A. japonicus*, *E. fraudatrix* synthesizes parkeol sterols and lanostadienol glycosides).

### 2.4. Gene Structure Determination and Analysis of OSC Genes

The *Ef*OSC1 and *Ef*OSC2 encoding gene sequences were obtained by a series of PCRs using gene-specific primers designed based on the cDNA sequences. As a result of PCR optimization, fragments ranging from 4 to 18 kb were obtained. The resulting amplicons were sequenced using Oxford Nanopore Technologies, assembled using CLC Genomic Workbench 24.0.1, and manually corrected. The final OSCs gene sequences had a length of about 30–40 kb for *OSC1* and 22–25 kb for *OSC2*. In addition, OSC1 and OSC2 cDNA sequences were mapped on the *E. fraudatrix* draft genome using BLAST, and the regions with the best hits were extracted. A total of four sequences were taken for each OSC gene (two from the amplicons assembly and two from the draft genome sequence), which were used for comparison with cDNA sequences and further haplotype analysis. After alignment with cDNA sequences, it became clear that the range of gene lengths was due to significant differences in intron sizes. The exon–intron structures of *E. fraudatrix* OSCs genes comprised 19 exons and 18 introns (Figure 4a), with conservation of the GT/AG splice sites in both genes. The *OSC1* 5′UTR was located in exon 1, while the *OSC2* 5′UTR occupied exon 1 and partially occupied exon 2. The last codons of CDSs and 3′ UTRs were aligned on the same 19th exon for both genes. The lengths of the corresponding protein-coding exons of *OSC1* and *OSC2* were equal, whereas the lengths of the corresponding introns varied significantly and ranged from 0.4 to 7.9 kb. Moreover, the lengths of some introns varied among haplotypes; for example, the size of *OSC1* intron 2 ranged from 1.1 to 4.4 kb (Figure 4a).

Although the OSC1 and OSC2 transcript sequences show a high degree of similarity (Figure 4b, **1**), alignment of the gene sequences using NCBI BLAST Global Alignment revealed an almost complete absence of aligned regions (Figure 4b, **3**), indicating an exceptionally high degree of intronic divergence between *E. fraudatrix* OSC1 and OSC2 genes. This suggests a significant evolutionary rate for intronic regions while preserving overall sequence similarity at the transcript and protein levels (Figure 4b, **1**,**2**). When *OSC1* and *OSC2* haplotypes were aligned (Figure 4c,d), a significant degree of polymorphism was found in *OSC1* introns, whereas *OSC2* introns showed greater conservation. The high polymorphism and larger size of the OSC1 gene suggest that this gene may be an ancestor of the OSC2 gene.

## 3. Materials and Methods

### 3.1. Sample Collection and Nucleic Acids Extraction

The samples of sea cucumber *E. fraudatrix* were collected in Troitsa and Sobol Bays (Peter the Great Gulf, the Sea of Japan) during 2019–2024. Species identification was carried out by Dr. I.Y. Dolmatov (A.V. Zhirmunsky National Scientific Center of Marine Biology, FEBRAS, Vladivostok, Russia). Genomic DNAs and total RNAs were extracted from frozen or fresh *E. fraudatrix* tissues (body wall, muscle, gut) using TRIzol reagent (Thermo Fisher Scientific, Waltham, MA, USA) according to the manufacturer’s manual, except that the final RNA elution step was performed with HiDi Formamide (Thermo Fisher Scientific, Waltham, MA, USA) and stored in −70 °C. Nucleic acids concentration and purity were determined using a NanoPhotometer P330 (Implen, Munich, Germany), and their integrity was assessed using electrophoresis in 1.2% agarose gel.

This study was carried out in accordance with the recommendations of the Convention on Biological Diversity and was approved by the Ethics Committee of the G.B. Elyakov Pacific Institute of Bioorganic Chemistry, FEBRAS, Vladivostok, Russia (Protocol No. 0037; 12 March 2021).

### 3.2. cDNA Synthesis, RACE, and PCR Amplification

The synthesis of full-length-enriched double-stranded cDNAs was carried out using total RNAs, pretreated with DNase I (Thermo Fisher Scientific, Waltham, MA, USA) and the MINT Universal reagent kit (Evrogen, Moscow, Russia) in accordance with the manufacturer’s protocols. The 3′- and 5′-flanking sequences of OSC cDNAs were obtained by the rapid amplification of cDNA ends (RACE) technique using the RACE primer set (Evrogen, Moscow, Russia) and gene-specific primers (Table 2). The primers were designed based on partial *OSC* sequences from *E. fraudatrix* transcriptome GHCL00000000 [15]. The RACE amplifications were conducted with Encyclo^®^ DNA Polymerase (Evrogen, Moscow, Russia) according to the scheme in Appendix A. RACE-PCR-fragments were cloned and sequenced using an ABI 3130xl Genetic Analyzer (Applied Biosystems, Hitachi, Tokyo, Japan). The cDNA sequences were verified by PCR amplification with OSC1_start/OSC1_stop and OSC2_start/OSC2_stop primers (Table 2) using Q5^®^ High-Fidelity DNA Polymerase (New England Biolabs, Ipswich, MA, USA) under the following temperature conditions: 98 °C for 30 s, 35 cycles of 98 °C for 10 s, 62 °C for 15 s, 72 °C for 70 s, and, finally, 72 °C for 300 s. Obtained full-length CDSs were cloned using the InsTAclone PCR Cloning Kit (Thermo Fisher Scientific, Waltham, MA, USA), and positive clones were sequenced with gene-specific primers using SeqStudio™ Genetic Analyzer (Thermo Fisher Scientific, Waltham, MA, USA). The obtained sequences were deposited into GenBank databases under the accession numbers OR711403.1 and OR725688.1, respectively, for *OSC1* and *OSC2*.

### 3.3. Gene Structure Determination

Amplification of OSC1 and OSC2 gene fragments from genomic DNAs with sizes between 5 and 12 Kbp was performed with a Long PCR Enzyme Mix (Thermo Fisher, Waltham, MA, USA) and gene-specific primers (Table 2) under the following temperature conditions: 94 °C for 1 min, 30 cycles of 94 °C for 20 s, 58 °C for 30 s, and 68 °C for 600 s. Fragments with sizes larger than 12 Kbp were obtained with a KOD One Master Mix (Toyobo, Osaka, Japan) according to the manufacturer’s protocol. The ONT DNA libraries were prepared using SQK-NBD114.24 (Oxford Nanopore Technologies, Oxford, UK) and sequenced with MinION Mk1b on flow cell FLO-MIN114 (Oxford Nanopore Technologies, Oxford, UK). The ONT raw reads were basecalled with Dorado v.0.7.1 (Oxford Nanopore Technologies, Oxford, UK). The BRAKER version 3.0.6 [29] was used to predict de novo OSC genes and untranslated regions (UTRs) using both RACE-PCR and transcriptomic data of *E. fraudatrix* [15], the protein dataset of Metazoa from OrthoDB version 11 [30], and proteins of *Apostichopus japonicus* [31], *Holothuria leucospilota* [32], *Acanthaster planci* (GCF_001949145.1) [33], *Asterias rubens* (GCF_902459465.1), *Strongylocentrotus purpuratus* (GCF_000002235.5) [34], *Lytechinus variegatus* [35], and *Patiria miniata* (GCF_015706575.1) uploaded from NCBI [36] and Echinobase [37]. Proteins containing sequences were extracted and processed through QIAGEN CLC Genomic Workbench, version 24.0.1 (QIAGEN, Aarhus, Denmark).

### 3.4. Sequence Analysis, Homology Modeling, and Docking

Multiple sequence alignments of OSCs were performed using ClustalW implemented in MEGA 11, version 11.0.13 [38], or QIAGEN CLC Genomic Workbench, version 24.0.1 (QIAGEN, Aarhus, Denmark), and manually corrected. A maximum-likelihood (ML) phylogeny with 500 nonparametric bootstrap replicates was obtained using an LG+G substitution model calculated in MEGA 11, version 11.0.13 [38].

Comparisons of OSCs transcripts sequences obtained from cDNA and genes haplotypes assembled after genome and amplicons sequencing with the resulting scatter plots were performed using NCBI BLAST Global Alignment with default options [39].

Theoretical models of *Ef*OSC1, *Ef*OSC2, *Aj*LAS1, and *Aj*LAS2 were obtained using the MOE package, version 2018.01 (Molecular Operating Environment; Chemical Computing Group ULC, 1010 Sherbrooke St. West, Suite #910, Montreal, QC, Canada, H3A 2R7, 2018). For molecular docking, the models of OSCs complexes with parkeol or lanostadienol were optimized by forcefield Amber10:EHT, using the crystal structure of human OSC in complex with lanosterol (PDB code 1W6K) [26] as a template after removing water molecules. The MOE docking parameters were set using default values. Contact analysis and visualization of the results were carried out using Ligand interaction and Dock modules in the MOE 2018.01 program and Discovery Studio Visualizer v24.1.0.23298 (Dassault Systèmes: San Diego, CA, USA, 2024). Structure alignments were performed using QIAGEN CLC Genomic Workbench, version 24.0.1 (QIAGEN, Aarhus, Denmark).

## 4. Conclusions

Our studies indicate that sea cucumber *E. fraudatrix* (family Sclerodactylidae, order Dendrochirotida) has two OSC genes, as well as representatives of the genera *Apostichopus* and *Stichopus* (family Stichopodidae). Based on the results of sequence analysis and molecular docking, we suggest that OSC1 with residue L436 seems to be parkeol synthase, while OSC2 with residue Q439 appears to be lanostadienol synthase, and the above allows us to assume that these residues are key in identifying orthologous groups. We believe that if *E. fraudatrix* OSCs form their own separate phylogenetic clade, then this evolutionary divergence between holothurians of the Sclerodactylidae and Stichopodidae families was probably driven by differences in the use of triterpene scaffolds for sterol and glycoside biosynthesis. Considering the transcript and gene structures of *E. fraudatrix OSCs* haplotypes, we suggest that the OSC1 gene can be an ancestor of the OSC2 gene. Taken together, these findings, firstly acquired for the Sclerodactylidae family (order Dendrochirotida), bring some additional clarity regarding the origin and biosynthesis of triterpenoid glycosides in sea cucumbers.

## Figures and Tables

**Figure 1 ijms-25-12881-f001:**
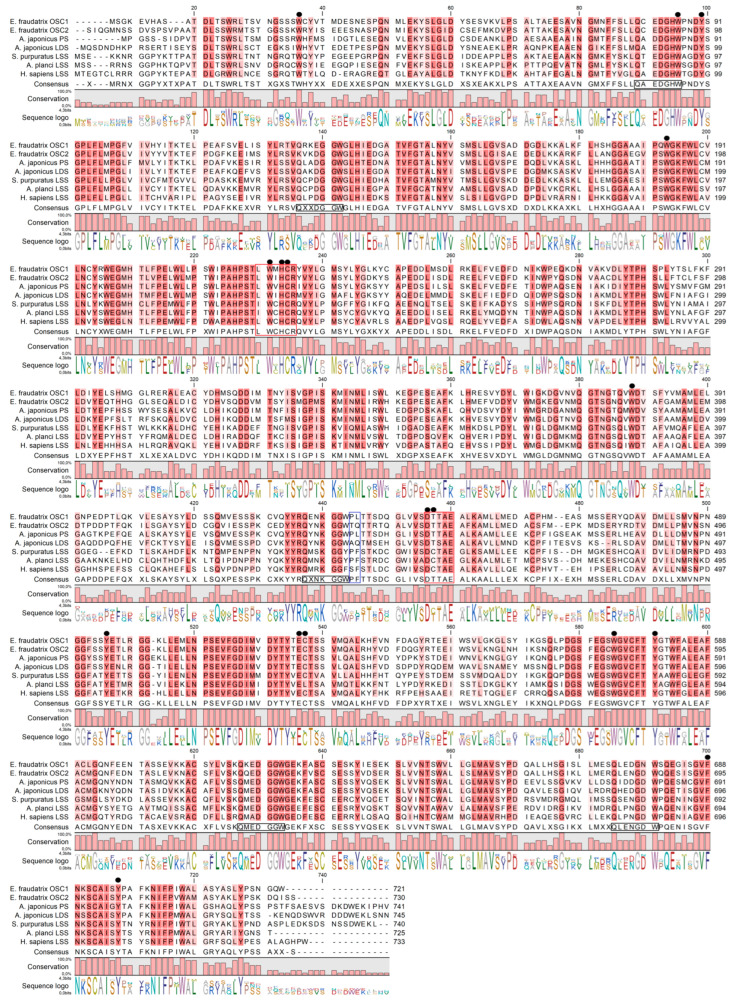
Alignment of OSCs proteins from sea cucumbers, sea urchin, starfish, and human. *E. fraudatrix* OSC1 and OSC2 (OR725688 and OR711403 in this study); *A. japonicus* LAS1 (PS) and LAS2 (LDS) (ON478352.1 and ON478353.1, respectively); *A. parvimensis* PS and LDS (ON478351.1 and ON478350.1); *S. horrens* OSC1 and OSC2 [14]; LSS: *A. planci* (XM 022227483.1), *S. purpuratus* (ON478349.1), *H. sapiens* (NM 002340.6). Conservative residues are in red colors. Active site residues are indicated by black circles. QXXXW motifs are marked on a consensus sequence by black boxes. LWIHCR and DTTAE motifs are marked by red boxes. Key residues, which determine enzyme function, are labeled with the blue box.

**Figure 2 ijms-25-12881-f002:**
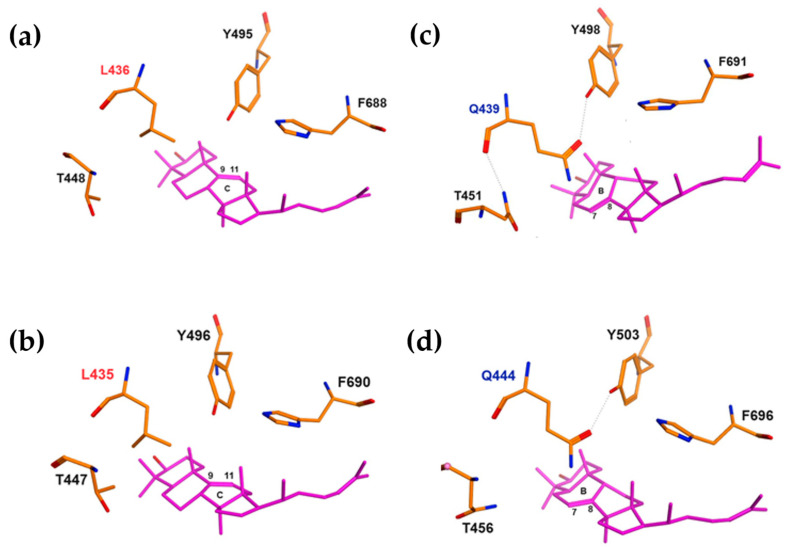
Molecular docking of triterpenoid in the active site of oxidosqualene cyclase. Functionally significant residues L436/L435 of *Ef*OSC1 (**a**) and *Aj*LAS1 (**b**) near the B and C rings of parkeol are shown in red. Functionally significant residues Q439/Q444 of *Ef*OSC2 (**c**) and *Aj*LAS2 (**d**) near the B ring of lanostadienol are shown by blue.

**Figure 3 ijms-25-12881-f003:**
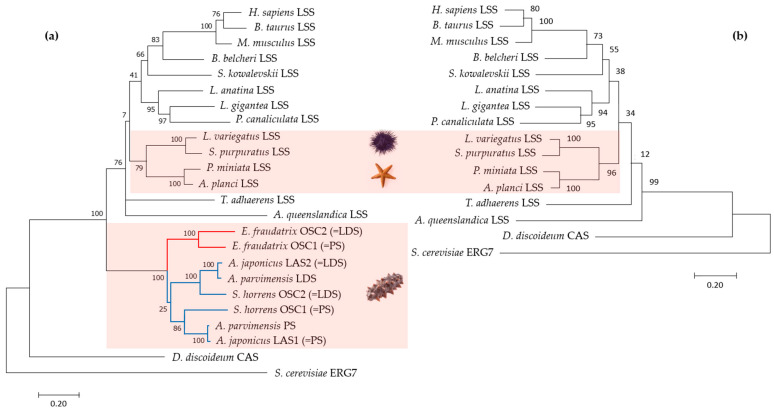
Maximum likelihood (ML) phylogenetic trees based on amino acid sequences of OSCs with (**a**) and without (**b**) sea cucumbers: *H. sapiens* (NM 002340.6), *B. taurus* (NM 001046564.1), *M. musculus* (XM 036155651.1), *B. belcheri* (XM 019790446.1), *S. kowalevskii* (XM 006825036.1), *L. anatina* (XM 013562424.1), *L. gigantea* (XM 009046756.1), *P. canaliculata* (XM 025240299.1), *L. variegatus* (XM 041622296.1), *S. purpuratus* (ON478349.1), *P. miniata* (ON478348.1), *A. planci* (XM 022227483.1), *T. adhaerens* (XM 002110738.1), *A. queenslandica* (XM 003383129.3), *E. fraudatrix* OSC1 (OR725688, this study), *E. fraudatrix* OSC2 (OR711403, this study), *A. japonicus* LAS1 (ON478352.1), *A. japonicus* LAS2 (ON478353.1), *A. parvimensis* PS (ON478351.1), *A. parvimensis* LDS (ON478350.1), *S. horrens* OSC1 [14], *S. horrens* OSC2 [14], *D. discoideum* (XM 641154.1), and *S. cerevisiae* (NP 011939.2).

**Figure 4 ijms-25-12881-f004:**
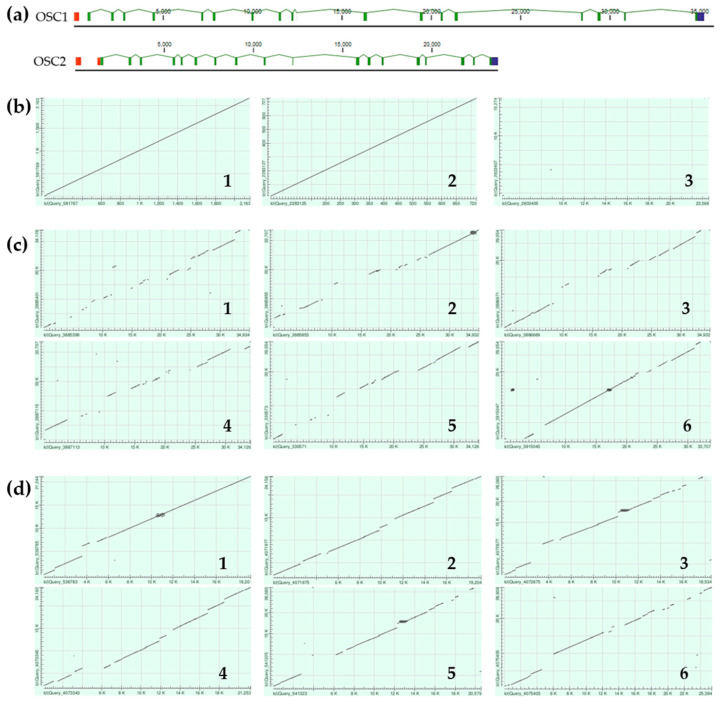
Comparison of *E. fraudatrix* OSCs gene structures and their haplotypes. (**a**) Gene structure schemes. (**b**) Scatter plots of OSC1 and OSC2 transcripts (**1**), deduced amino acid sequences (**2**), and gene sequences (**3**) comparison. (**c**) Scatter plots of pairwise alignment through BLAST of four OSC1 gene haplotypes (comparison **1**—haplotypes 1 and 2; **2**—haplotypes 1 and 3; **3**—haplotypes 1 and 4; **4**—haplotypes 2 and 3; **5**—haplotypes 2 and 4; **6**—haplotypes 3 and 4). (**d**) Scatter plots of pairwise alignment through BLAST of four OSC2 gene haplotypes (comparison **1**—haplotypes 1 and 2; **2**—haplotypes 1 and 3; **3**—haplotypes 1 and 4; **4**—haplotypes 2 and 3; **5**—haplotypes 2 and 4; **6**—haplotypes 3 and 4).

**Table 1 ijms-25-12881-t001:** Amino acid sequence identity (bottom left) and similarity (top right) between holothurian OSCs and LSSs from other Echinoderms.

	OSC1	OSC2	*Aj*LAS1	*Aj*LAS2	*Ap*PS	*Ap*LDS	*Sh*OSC1	*Sh*OSC2	*Sp*LSS	*Ap*LSS
*E. fraudatrix* OSC1	100	87.18	82.11	81.97	82.66	81.69	82.25	81.84	73.38	72.65
*E. fraudatrix* OSC2	77.5	100	81.17	80.62	81.31	80.76	79.44	80.85	71.89	71.81
*A. japonicus* LAS1 *	70.6	67.2	100	84.86	98.79	84.89	88.48	83.47	72.73	73.71
*A. japonicus* LAS2 *	69.9	67.3	72.4	100	85.27	98.27	83.83	90.21	73.00	75.71
*A. parvimensis* PS *	71.0	67.5	98.3	72.4	100	85.30	88.31	83.62	72.86	74.02
*A. parvimensis* LDS *	69.7	67.8	72.4	96.0	73.1	100	83.54	90.62	72.73	75.14
*S. horrens* OSC1	68.7	64.7	76.6	68.5	76.1	69.1	100	86.02	73.52	74.75
*S. horrens* OSC2	68.1	67.5	71.3	80.1	71.6	81.3	71.6	100	73.68	75.32
*S. purpuratus* LSS *	57.75	55.65	58.89	57.30	58.75	56.75	57.46	57.52	100	80.77
*A. planci* LSS *	57.92	56.37	59.11	58.19	58.94	57.63	57.55	57.97	69.43	100

* PS = LAS1—parkeol synthase, LDS = LAS2—lanostadienol synthase, LSS—lanosterol synthase. *E. fraudatrix* OSC1 and OSC2 (OR725688 and OR711403 in this study); *A. japonicus (Aj)* LAS1 (PS) and LAS2 (LDS) (ON478352.1 and ON478353.1, respectively); *A. parvimensis (Ap)* PS and LDS (ON478351.1 and ON478350.1); *S. horrens (Sh)* OSC1 and OSC2 [12]; LSS: *A. planci (Ap)* (XM 022227483.1), *S. purpuratus (Sp)* (ON478349.1).

**Table 2 ijms-25-12881-t002:** Primers used in this study.

Primers Name	5′ -> 3′ Sequence	Objectives
OSC1_start	ATGTCTGGCAAAGAKGTSCATGCAAGTG	OSC1 cDNA amplification and gDNA amplicons sequencing.
OSC1_stop	ATCACTGGCCATTTGAAGGATAGAGCCTG	
OSC2_start	AGAATTCAAGGAATGAATTCCAGTGACGTCTC	OSC2 cDNA amplification and gDNA amplicons sequencing.
OSC2_stop	TCTCGAGATCTGGTCCTTCGATGGGTAGAGCT	
OSS_838For	GCCTACTCATACCTTGATTCTTCACAGA	OSC1 RACE and gDNA amplicons sequencing.
OSS_865Rev	TCTGTGAAGAATCAAGGTATGAGTAGGC	
OSC2_F2	AGGACCAAAGAGATAATGTTGCTGCATGT	OSC2 RACE and gDNA amplicons sequencing.
OSC2_2RR	TGTTGGATCATCGGGCGTAT	
OSS_1143Rev	CAACATCTCCAAGAGTTTGCCWCCTCGG	OSC1 RACE and Control PCR of gDNA with OSS_838F.

## Data Availability

The assembled haplotypes of OSC1 and OSC2 genes and protein alignment files, as well as the Excel table of enzyme–ligand interaction distances and the PDB file with structural alignment can be accessed through the Figshare with the provided link (https://doi.org/10.6084/m9.figshare.26242250).

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
