# Peer review of "Structure of Genes Encoding Oxidosqualene Cyclases—Key Enzymes of Triterpenoid Biosynthesis from Sea Cucumber Eupentacta fraudatrix"

_ijms, 2024, doi:10.3390/ijms252312881_

Round 1

Reviewer 1 Report

Comments and Suggestions for Authors

This paper focused on identifying and characterizing Oxidosqualene Cyclases (OSCs) in the sea cucumber. By doing that, two OSC gene candidates, OSC1 and OSC2, were identified, showing high sequence identity to each other and predicting parkeol synthase and lanostadienol synthase activities, respectively. The phylogenetic analysis revealed their relationship with other holothurian OSCs. This study highlighted the genetic complexity and evolutionary pathways of OSCs in E. fraudatrix, contributing to the broader understanding of triterpene biosynthesis in marine organisms. Although the results themselves are excellent, analyses and discussion of the obtained results and the writing itself need to be improved on, as well as some minor points that are at the authors' discretion.

1. Line 40, The sentence "It has been found that in sea cucumber E. fraudatrix all free, resulted by de novo biosynthesis," contains an error. It appears that there may have been a mistranslation or typo in this part of the sentence? The intended meaning of "all free, resulted by de novo biosynthesis" is not clear. It seems there may be missing or incorrect words.

2. Line 109, “Comparison of the primary structures of OSCs and LSSs from echinoderms (Table 2)”, here Table 2 should be Table 1. And Line 131, the same mistake.

3. Line 150, the reference cited should be corrected to [10] instead of [11].

4. Figure 4b, c and d, how do the authors perform this analysis? Please add a detailed description in the section of Methods. How did the authors identify two sequences for each OSC gene in the draft genome? Were these sequences derived from parental genomes? Standard genome sequencing methods cannot differentiate between parental sequences. Did the authors perform haploid genome sequencing, or were the authors able to detect two OSC1 genes and two OSC2 genes?

Author Response

Response to Reviewer 1:

Thank you very much for taking the time to review our manuscript. We appreciate your helpful comments and suggestions. We have revised our manuscript in accordance with your recommendations.

Comment 1: Line 40, The sentence "It has been found that in sea cucumber E. fraudatrix all free, resulted by de novo biosynthesis," contains an error. It appears that there may have been a mistranslation or typo in this part of the sentence? The intended meaning of "all free, resulted by de novo biosynthesis" is not clear. It seems there may be missing or incorrect words.

Response: Thank you for attentiveness, it’s missing “sterols” after “all free”. It was corrected to “It has been found that in sea cucumber E. fraudatrix all free sterols, resulted by de novo biosynthesis”. Line

Comment 2: Line 109, “Comparison of the primary structures of OSCs and LSSs from echinoderms (Table 2)”, here Table 2 should be Table 1. And Line 131, the same mistake.

Response: Thank you, the corrections have been made.

Comment 3: Line 150, the reference cited should be corrected to [10] instead of [11].

Response: Reference [Li et al., ref 11] (now reference __) is used here correctly as this work was published in 2018 and contains data of LAS1 and LAS2 product determination by yeast expression. Please see at Li et al. 2018 (Results “Saponin biosynthesis and convergent evolution”), where it is clearly visible in Figure 3d.

Comment 4: Figure 4b, c and d, how do the authors perform this analysis? Please add a detailed description in the section of Methods. How did the authors identify two sequences for each OSC gene in the draft genome? Were these sequences derived from parental genomes? Standard genome sequencing methods cannot differentiate between parental sequences. Did the authors perform haploid genome sequencing, or were the authors able to detect two OSC1 genes and two OSC2 genes?

Response: Detailed description of the analysis to produce Figure 4b, c and d has been added to the Materials and Methods section, Sequence analysis, homology modeling and docking. We consider that the use of the Oxford Nanopore MinION platform provides an overlap of sufficient length to resolve haplotypes of genes with significant differences in introns. In case where introns not variate by size, standard sequencing methods cannot provide resolution between parental sequences, as you said in your comment. We showed in our manuscript, the differences in intron sizes are significant for both OSC1 and OSC2. It should be clarified that sequencing of the draft genome was performed with Oxford Nanopore Technology. We then do not need to perform haploid genome sequencing, and we easily found two OSC1 genes and two OSC2 genes by performing rough assembly annotation using BRAKER version 3.0.6.

Reviewer 2 Report

Comments and Suggestions for Authors

This study investigates oxidosqualene cyclases (OSCs) in Eupentacta fraudatrix, a sea cucumber species known for its distinctive triterpenoid biosynthesis. The researchers employed molecular and bioinformatics approaches to identify and characterize two OSC genes (OSC1 and OSC2), which encode enzymes likely involved in catalyzing the production of specific triterpenoids. Further analysis predicted specific residues (L436 in OSC1 and Q439 in OSC2) that are thought to determine product specificity. Phylogenetic analysis revealed that the OSC genes in E. fraudatrix form a distinct branch, diverging from OSCs in closely related sea cucumber species, likely due to evolutionary divergence. These findings provide insights into the genetic diversity of OSC genes in sea cucumbers and their evolutionary significance. However, some conclusions require further validation and remain open to discussion. Therefore, I recommend the paper be accepted after revision.

Major Concerns:

1. The overall data structure is convincing; however, biochemical validation of the predicted enzyme activities is lacking.

2. The conclusion regarding the uniqueness of the E. fraudatrix OSC genes' evolutionary pathway is not fully solid. Where is their uniqueness reflected? The two OSC enzymes produce different products, yet they do not cluster with corresponding enzyme types in the phylogenetic tree (particularly among sea cucumbers). Why is this? There is still a lack of supporting evidence and sufficient discussion.

3. Relying solely on molecular docking and sequence analyses does not sufficiently demonstrate the decisive role of L436 and Q439 in determining catalytic products. It would be beneficial to include additional experiments or data, or to provide thorough literature support and discussion to reinforce this conclusion.

4. The manuscript could better clarify and highlight the importance and significance of this research, such as emphasizing the role of OSC products in sea cucumber immunity and their potential applications in medicine.

Minor Points:

1. In Table 1, it would be helpful to include brief explanations of abbreviations such as LAS, LSD, and LSS in the header for easier reading. Additionally, are AjLAS1 and AjLAS2 in Table 1 consistent with the description in the Figure 1 legend, "A. japonicus LAS1 (=PS) and LAS2 (=LDS)"? Could the related annotations be standardized throughout the text?

2. In Figure 1, the critical residues L436 and Q439 are not highlighted.

3. The resolution of Figure 4 is low, making many data details difficult to see clearly. Additionally, the meanings of the numbers 4, 5, and 6 are not explained in the figure legend.

4. Has the expression of OSCs been verified in various tissues of E. fraudatrix? Are there any differences in their tissue distribution and expression?

Author Response

Response to Reviewer 2:

Thank you very much for taking the time to review our manuscript. We appreciate your helpful comments and suggestions. We have revised our manuscript in accordance with your recommendations.

Comment 1: The overall data structure is convincing; however, biochemical validation of the predicted enzyme activities is lacking.

Response: Thank you for your advice. We plan to carry out biochemical validation of the predicted enzyme activities using yeast heterologous expression.

Comment 2: The conclusion regarding the uniqueness of the E. fraudatrix OSC genes' evolutionary pathway is not fully solid. Where is their uniqueness reflected? The two OSC enzymes produce different products, yet they do not cluster with corresponding enzyme types in the phylogenetic tree (particularly among sea cucumbers). Why is this? There is still a lack of supporting evidence and sufficient discussion.

Response: The uniqueness lies in the presence of two enzymes in sea cucumbers, and not just one like in other animals. Second, the uniqueness of this phenomenon is that orthologs within a class (holothurians) are not grouped together; paralogs are closer to each other in terms of the degree of similarity.

Comment 3: Relying solely on molecular docking and sequence analyses does not sufficiently demonstrate the decisive role of L436 and Q439 in determining catalytic products. It would be beneficial to include additional experiments or data, or to provide thorough literature support and discussion to reinforce this conclusion.

Response: We presented literature support by the work of Thimmappa et al. 2022, where they provided sufficient evidence that replacing just one F residue with L or Q in the 440 position of LSS of the sea urchin S. purpuratus resulted in the production of detectable levels of parkol or lanostadienol, respectively (along with lanosterol). Unfortunately, there is no other data.

Comment 4: The manuscript could better clarify and highlight the importance and significance of this research, such as emphasizing the role of OSC products in sea cucumber immunity and their potential applications in medicine.

Response: We have added additional information in the Introduction section to highlight the potential applications of E. fraudatrix OSC end products in biomedicine.

Comment 5: In Table 1, it would be helpful to include brief explanations of abbreviations such as LAS, LSD, and LSS in the header for easier reading. Additionally, are AjLAS1 and AjLAS2 in Table 1 consistent with the description in the Figure 1 legend, "A. japonicus LAS1 (=PS) and LAS2 (=LDS)"? Could the related annotations be standardized throughout the text?

Response: Thank you very much. A footer has been added above Table 1. A. japonicus LAS1 = A. japonicus PS, A. japonicus LAS2 = A. japonicus LDS. Various enzyme names arose from discussion of references [10] or [11].

Comment 6: In Figure 1, the critical residues L436 and Q439 are not highlighted.

Response: The critical residues were marked with a blue box, and a corresponding comment was provided in the description of Figure 1.

Comment 7: The resolution of Figure 4 is low, making many data details difficult to see clearly. Additionally, the meanings of the numbers “4,” “5,” and “6” are not explained in the figure legend.

Response: Description for Figure 4 has been made more detailed.

Comment 8: Has the expression of OSCs been verified in various tissues of E. fraudatrix? Are there any differences in their tissue distribution and expression?

Response 8: Thank you for your question. We have added more details to Results and Discussion (2.1.1) with tissue-specific product amplification, picture of electrophoresis was added in Supplementary materials.

Reviewer 3 Report

Comments and Suggestions for Authors

This manuscript represents a significant contribution to marine natural products research, particularly to the understanding of triterpenoid biosynthesis in sea cucumbers. The integration of molecular biology, structural biochemistry, and phylogenetics is a notable strength. Addressing limitations by experimentally validating the results, expanding the comparative analyses, and improving the discussion would greatly increase the scientific impact and accessibility of the manuscript.

In particular, the study provides critical insights into triterpenoid biosynthesis pathways in Eupentacta fraudatrix, a sea cucumber species from the family Sclerodactylidae. It is the first to characterize two oxidosqualene cyclase (OSC) genes, OSC1 and OSC2, broadening the understanding of sterol and triterpenoid metabolism in echinoderms.

The authors use advanced techniques such as RACE-PCR, Oxford Nanopore Technologies, molecular docking and phylogenetic analyses. This approach allows for in-depth analysis of OSC gene structure, sequence divergence and enzyme function.

The study highlights the evolutionary divergence of OSC genes in E. fraudatrix compared to other sea cucumbers, suggesting an ancestral relationship between OSC1 and OSC2. This contributes to a broader understanding of gene duplication and functional specialization in marine organisms.

Molecular docking and structural modeling analyses provide strong evidence that OSC1 functions as a parkeol synthase, while OSC2 acts as a lanostadienol synthase. Analysis of key residues (e.g. L436 and Q439) strengthens the biochemical interpretation of the enzyme specificity.

The results have implications for the study of bioactive compounds, especially triterpene glycosides, known for their pharmacological properties. The study opens new perspectives for research on the applications of these metabolites in the medical and biotechnological fields.

But I note some Weaknesses:

The functional predictions of OSC1 and OSC2 as parkeol and lanostadienol synthase are mainly based on molecular docking and sequence analyses. Experimental validation, e.g. by enzymatic assays or heterologous expression in yeast, would have provided more conclusive evidence.

Although the study compares OSCs from E. fraudatrix with those from Stichopodidae, a broader phylogenetic analysis including more distant echinoderms or other metazoans would have strengthened the evolutionary conclusions.

The potential ecological or physiological roles of the OSCs identified in E. fraudatrix are not discussed in detail. Insights into how these enzymes contribute to the organism's adaptation or defense mechanisms would improve the impact of the study.

The phylogenetic trees and docking analysis figures could benefit from additional annotations and detailed explanations of statistical metrics (e.g. bootstrap values). This would improve the accessibility and interpretability of the data.

Terms such as "QW motifs," "DTTAE domain," and "homology modeling" are not sufficiently explained, risking alienating readers who are not specialists in structural biochemistry or molecular docking.

I propose both general improvements, which concern the research and some specific ones, which concern the manuscript.

Specific:

Currently, the conclusions on the functions of OSC1 and OSC2 as parkeol and lanostadienol synthase are based on computational analyses. To make the results more robust, the results of enzymatic assays that test the catalytic activity of OSC1 and OSC2 proteins in vitro using substrates such as oxidosqualene should be provided. These experiments would provide direct confirmation of the specificity of the product.

Expand the Comparative Phylogenetic Analysis, in fact, the current analysis focuses mainly on sea cucumbers of the Stichopodidae and Sclerodactylidae families. To strengthen the evolutionary conclusions, one could:

Include data from other echinoderms: Integrate OSC sequences from starfish, sea urchins and other marine organisms to better delineate the evolutionary relationships among echinoderms.

The OSC sequences of sea cucumbers could be compared with those of other metazoans, such as molluscs and annelids, to identify potential common or divergent evolutionary trajectories.

The biological implications of OSCs in E. fraudatrix deserve further investigation, such as assessing ecological adaptations by investigating how OSC1 and OSC2 contribute to the species' ability to withstand extreme environmental conditions or predators.

Testing the role of parkeol and lanostadienol in chemical defense mechanisms against predators or pathogens.

Evaluating whether differences in OSC profiles between the families Sclerodactylidae and Stichopodidae are influenced by different habitats or selective pressures related to the environment.

Improving the visualization of current figures that, although detailed, could be made more accessible and intuitive.

Phylogenetic trees could include more detailed annotations, such as well-visible bootstrap values, and adding markers that highlight key differences between clades.

Clarify and Expand Terminology for some technical terms that require additional clarity to ensure manuscript accessibility. Suggest Insert a section that briefly explains technical terms such as "QW motifs," "DTTAE domain," and "homology modeling."

Provide a more detailed description of techniques used, such as molecular docking and phylogenetic analysis

General:

Evaluate heterologous expression by inserting the OSC1 and OSC2 genes into a model system (e.g., yeast lacking lanosterol synthase) to observe direct production of parkeol or lanostadienol.

Evaluate site-directed mutagenesis by modifying key residues (e.g., L436 and Q439) and evaluate the effect on synthesized products. This may confirm the role of these residues in enzyme specificity.

Methodological differences between OSC studies may introduce bias.

We recommend:

Collaborate with other research groups to standardize data collection and analysis methods (e.g., bioinformatics tools and molecular docking conditions) to allow direct comparisons between species.

Re-evaluate old datasets using the advanced technologies described in this study to analyze pre-existing data from other species, improving consistency between studies.

To better understand evolutionary and functional dynamics, we recommend performing repeated sampling campaigns at different times of the year and in distinct habitats to observe seasonal or environmental variations in OSC profiles.

Finally, assess whether factors such as temperature, salinity or contamination influence OSC gene expression.

1. Abstract

Original:
"Oxidosqualene cyclases (OSCs), enzymes are responsible for converting linear triterpenes into tetracyclic ones, which known as precursors of other important and bioactive metabolites."
Correction:
"Oxidosqualene cyclases (OSCs) are enzymes responsible for converting linear triterpenes into tetracyclic ones, which are known as precursors of other important and bioactive metabolites."

Original:
"These data expand our knowledge of structural diversity and evolutionary changes in genes involved in triterpenoid biosynthesis in sea cucumbers."
Correction:
"These data expand our knowledge of the structural diversity and evolutionary changes in genes involved in triterpenoid biosynthesis in sea cucumbers."

2. Introduction

Original:
"Herein, we present the results of the transcript and gene structures determination of two OSCs from E. fraudatrix, whose deduced amino acid sequences showed notable difference when compared with OSCs from Stichopodidae."
Correction:
"Herein, we present the results of the transcript and gene structure determination of two OSCs from E. fraudatrix, whose deduced amino acid sequences showed notable differences compared to OSCs from Stichopodidae."

Original:
"These results show that gene duplication followed by neofunctionalization could have a significant role in the evolution of triterpenoid biosynthesis in holothurians."
Correction:
"These results suggest that gene duplication followed by neofunctionalization may play a significant role in the evolution of triterpenoid biosynthesis in holothurians."

3. Materials and Methods

Original:
"Total RNA was extracted using the RNeasy Mini Kit, according to the manufacturer's protocol."
Correction:
"Total RNA was extracted using the RNeasy Mini Kit following the manufacturer's protocol."

Original:
"Alignment of sequences were performed using Clustal Omega."
Correction:
"Alignment of sequences was performed using Clustal Omega."

4. Results

Original:
"The OSC1 and OSC2 gene structures include several exons and introns with distinct intronic regions, which potentially indicate differential regulatory mechanisms."
Correction:
"The OSC1 and OSC2 gene structures include several exons and introns, with distinct intronic regions potentially indicating differential regulatory mechanisms."

Original:
"The phylogenetic analysis demonstrated that OSC1 and OSC2 formed a separate clade from other holothurians, suggesting significant evolutionary divergence."
Correction:
"Phylogenetic analysis demonstrated that OSC1 and OSC2 form a separate clade from other holothurians, suggesting significant evolutionary divergence."

Original:
"Docking analysis revealed that active site residues are responsible for substrate specificity, particularly in OSC1."
Correction:
"Docking analysis revealed that the active site residues are responsible for substrate specificity, particularly in OSC1."

5. Discussion

Original:
"The identified residues such as L436 and Q439 are crucial for OSC enzymatic activity, which highlights the specificity of parkeol production."
Correction:
"The identified residues, such as L436 and Q439, are crucial for OSC enzymatic activity, highlighting the specificity of parkeol production."

Original:
"Our data support the hypothesis that gene duplication events led to the functional diversification observed in OSC1 and OSC2."
Correction:
"Our data support the hypothesis that gene duplication events have led to the functional diversification observed in OSC1 and OSC2."

Original:
"Future studies could be focused on the functional characterization of OSCs using heterologous expression systems."
Correction:
"Future studies could focus on the functional characterization of OSCs using heterologous expression systems."

6. Conclusion

Original:
"In conclusion, this study reveals novel insights into the gene structures and potential functions of OSC1 and OSC2 in E. fraudatrix, contributing to the understanding of triterpenoid biosynthesis pathways in sea cucumbers."
Correction:
"In conclusion, this study provides novel insights into the gene structures and potential functions of OSC1 and OSC2 in E. fraudatrix, contributing to the understanding of triterpenoid biosynthesis pathways in sea cucumbers."

Original:
"These findings can be extended to further studies on marine-derived triterpenoids and their applications."
Correction:
"These findings could be extended to future studies on marine-derived triterpenoids and their applications."

Author Response

Response to Reviewer 3:

Thank you very much for taking the time to review our manuscript. We appreciate your helpful comments and suggestions. We have revised our manuscript in accordance with your recommendations.

Comment 1: The functional predictions of OSC1 and OSC2 as parkeol and lanostadienol synthase are mainly based on molecular docking and sequence analyses. Experimental validation, e.g. by enzymatic assays or heterologous expression in yeast, would have provided more conclusive evidence.

Response:. Thank you for your advice. We plan to carry out biochemical validation of the predicted enzyme activities using yeast heterologous expression.

Comment 2: Although the study compares OSCs from E. fraudatrix with those from Stichopodidae, a broader phylogenetic analysis including more distant echinoderms or other metazoans would have strengthened the evolutionary conclusions.

Response: It must be said that the sequences of more distant echinoderms (starfish and sea urchins) and other multicellular animals have already been included in the analysis. Please, see Table 1 and Figures 1 and 3. You can see the following list of organisms: L. variegatus (XM 041622296.1), S. purpuratus (ON478349.1), P. miniata (ON478348.1), A. planci (XM 022227483.1), including other metazoans are presented by H. sapiens (NM 002340.6), B. taurus (NM 001046564.1), M. musculus (XM 036155651.1), B. belcheri (XM 019790446.1), S. kowalevskii (XM 006825036.1), L. anatina (XM 013562424.1), L. gigantea (XM 009046756.1), P. canaliculata (XM 025240299.1), T. adhaerens (XM 002110738.1), A. queenslandica (XM 003383129.3).

Comment 3: The potential ecological or physiological roles of the OSCs identified in E. fraudatrix are not discussed in detail. Insights into how these enzymes contribute to the organism's adaptation or defense mechanisms would improve the impact of the study.

Response: We have added additional information in the Introduction section to highlight the potential applications of E. fraudatrix OSC end products in biomedicine.

Comment 4: The phylogenetic trees and docking analysis figures could benefit from additional annotations and detailed explanations of statistical metrics (e.g. bootstrap values). This would improve the accessibility and interpretability of the data.

Response: Bootstrap values are presented on phylogenetic tree (Figure 3), for structure alignments RMSD and TM-score values are presented on Table S1, characteristics of bond types between active center and ligands are described on Figure S4, values and distances obtained by docking can be found in Excel table “interaction distances” (available via link on Figshare https://doi.org/10.6084/m9.figshare.26242250. This link was remained in manuscript part Data Availability Statement).

Comment 5: Terms such as "QW motifs," "DTTAE domain," and "homology modeling" are not sufficiently explained, risking alienating readers who are not specialists in structural biochemistry or molecular docking.

Response: The terms used do not require explanation in the text. References are provided for this.

Comment 6: Currently, the conclusions on the functions of OSC1 and OSC2 as parkeol and lanostadienol synthase are based on computational analyses. To make the results more robust, the results of enzymatic assays that test the catalytic activity of OSC1 and OSC2 proteins in vitro using substrates such as oxidosqualene should be provided. These experiments would provide direct confirmation of the specificity of the product.

Response 6: Thank you for your advice, we plain do it in our further work.

Comment 7: Expand the Comparative Phylogenetic Analysis, in fact, the current analysis focuses mainly on sea cucumbers of the Stichopodidae and Sclerodactylidae families. To strengthen the evolutionary conclusions, one could:

Include data from other echinoderms: Integrate OSC sequences from starfish, sea urchins and other marine organisms to better delineate the evolutionary relationships among echinoderms.

The OSC sequences of sea cucumbers could be compared with those of other metazoans, such as molluscs and annelids, to identify potential common or divergent evolutionary trajectories.

Response: The phylogenetic tree includes following classes of metazoans: Holothuroidea, Echinoidea, Asteroidea, Demospongiae, Gastropoda, Enteropneusta, Leptocardii, Mammalia and Uniplacotomia. This tree was constructed using maximum likelihood (ML) method and have high bootstrap values on most nodes.

Comment 8: The biological implications of OSCs in E. fraudatrix deserve further investigation, such as assessing ecological adaptations by investigating how OSC1 and OSC2 contribute to the species' ability to withstand extreme environmental conditions or predators.

Response: We have expanded section Introduction with brief description of earlier work aimed at studying the cytotoxic and immunomodulatory activities of glycosides, isolated from E. fraudatrix by in vitro and in vivo experiments.

Comment 9: Testing the role of parkeol and lanostadienol in chemical defense mechanisms against predators or pathogens.

Response: Thank you for your advice, we plain do it in our further work.

Comment 10: Evaluating whether differences in OSC profiles between the families Sclerodactylidae and Stichopodidae are influenced by different habitats or selective pressures related to the environment.

Response: Thank you for your advice, we plain do it in our further work, investigation aim to evaluate dependences holothurians OSCs expression profiles of environment properties by Real-time PCR is within our area of interest too.

Comment 11: Improving the visualization of current figures that, although detailed, could be made more accessible and intuitive.

Response: Thank you for your advice, we plain do it in our further work

Comment 12: Phylogenetic trees could include more detailed annotations, such as well-visible bootstrap values, and adding markers that highlight key differences between clades.

Response 12: Our Figure 3 with phylogenetic tree contain as well-visible bootstrap values and key branches highlighted.

Comment 13: Clarify and Expand Terminology for some technical terms that require additional clarity to ensure manuscript accessibility. Suggest Insert a section that briefly explains technical terms such as "QW motifs," "DTTAE domain," and "homology modeling.

Response 13: The corresponded references are cited:

  • Sato, T.; Hoshino, T. Functional Analysis of the DXDDTA Motif in Squalene-Hopene Cyclase by Site-Directed Mutagenesis Experiments: Initiation Site of the Polycyclization Reaction and Stabilization Site of the Carbocation Intermediate of the Initially Cyclized A-Rin. Biotechnol. Biochem. 1999, 63, 2189–2198. https://doi.org/10.1271/bbb.63.2189
  • Poralla, K.; Hewelt, A.; Prestwich, G.D.; Abe, I.; Reipen, I.; Sprenger, G. A specific amino acid repeat in squalene and oxidosqualene cyclases. Trends Biochem. Sci. 1994, 19(4), 157–158. https://doi.org/10.1016/0968-0004(94)90276-3
  • Kushiro, T.; Shibuya, M.; Ebizuka, Y. β-Amyrin synthase: Cloning of oxidosqualene cyclase that catalyzes the formation of the most popular triterpene among higher plants. J. Biochem. 1998, 256, 238–244. https://doi.org/10.1046/j.1432-1327.1998.2560238.x
  • Godio, R.P.; Martín, J.F. Modified oxidosqualene cyclases in the formation of bioactive secondary metabolites: Biosynthesis of the antitumor clavaric acid. Fungal Genet. Biol. 2009, 46(3), 232–242. https://doi.org/10.1016/j.fgb.2008.12.002
  • Lin, Y.L.; Lee, Y.R.; Tsao, N.W.; Wang, S.Y.; Shaw, J.F.; Chu, F.H. Characterization of the 2,3-Oxidosqualene Cyclase Gene from Antrodia cinnamomea and Enhancement of Cytotoxic Triterpenoid Compound Production. Nat. Prod. 2015, 78(7), 1556–1562. https://doi.org/10.1021/acs.jnatprod.5b00020
  • Siedenburg, G.; Jendrossek, D. Squalene-Hopene Cyclases. Environ. Microbiol. 2011, 77(12), 3905–3915. https://doi.org/10.1128/AEM.00300-11
  • Corey, E.J.; Cheng, H.; Baker, C.H.; Matsuda, S.P.T.; Li, D.; Song, X. Studies on the Substrate Binding Segments and Catalytic Action of Lanosterol Synthase. Affinity Labeling with Carbocations Derived from Mechanism-Based Analogs of 2,3-Oxidosqualene and Site-Directed Mutagenesis Probes. Am. Chem. Soc. 1997, 119, 1289–1296. https://doi.org/10.1021/ja963228o
  • Chen, K.; Zhang, M.; Ye, M.; Qiao, X. Site-Directed Mutagenesis and Substrate Compatibility to Reveal the Structure–Function Relationships of Plant Oxidosqualene Cyclases. Prod. Rep. 2021, 38, 2261–2275. https://doi.org/10.1039/D1NP00015B.

Comment 14: Correction of language:

Response: Unfortunately, we were unable to find most sentences you recommended to correct.
